# Cross-National Active Learning in Global Development Studies: De-Colonizing the Curriculum

## Mary Jane Parmentier

School for the Future of Innovation in Society, Arizona State University, Tempe, AZ 85287, USA;
mj.parmentier@asu.edu

**Abstract:** De-colonizing the curriculum and active learning approaches that engage students experientially are both current themes in the teaching of International Studies and related disciplines. For the discipline of global development, both are critically needed approaches to training students who are able to work across national contexts and effectively interact with communities of different political histories and cultures. Yet neither is necessarily straightforward. This article explores two pedagogical projects that, while very different from each other, reveal commonalities through a technique of systemist notation and visualization, strengthening their contribution to cross-cultural and cross-national active learning. While online international collaborations and study abroad programs are different pedagogical contexts, they both involve significant levels of intercultural communication and knowledge exchange, neither of which is a given and requires careful course design and implementation.

**Keywords:** collaboration; culture; decolonizing; global development pedagogy; study abroad

## 1. Introduction

The imperative to de-colonize universities and curricula across the globe, but particularly in former colonial powers, has proliferated and become an ongoing discourse in academia over the past decade or more. Global development studies in particular, with a close connection to international development policies and practices, has mobilized to some extent in this de-colonizing endeavor, with scholars and students participating in reflexive exercises and dialogue, questioning many past assumptions (Ziai 2020; Sultana 2019). Stating one's 'positionality' has become a requirement in some research programs, whereby the researcher reflects and discloses their societal positionality vis-a-vis the targeted research community. It has become common to 'de-colonize' one's syllabus in any discipline by at least including scholars from the Global South. A recent Google Scholar search for 'decolonizing the syllabus' resulted in over ten pages of articles focused on decolonizing academia, including, at the center, decolonizing the syllabus. This high level of activity began in 2019 and included disciplines across the social sciences and humanities. A search for 'decolonizing global development studies' resulted in even more search pages full of recent articles as well as research going back a decade or more.

What these Google searches have in common is the confirmation that (a) knowledge production from the Global North proliferated during the era of imperialism and colonization; and (b) there is a need to actively recognize and expose the biases, as well as the damage done by the institution of colonialism and the legacies that remain (for example, Zidani 2021; Cornwall 2020; Patel 2020; Ziai 2020; Sultana 2019; Langdon 2013; Gorski 2008). It also should be noted that the majority of this literature comes from the United Kingdom, Canada, and, to a lesser extent, the United States. Communication primarily in English can therefore be observed as one lingering effect of the colonial era, even as the status quo is brought into question.

However, there does not appear to be a consensus on how we go about achieving the goal of decolonizing the curriculum, notably within global development studies. The

overall objective of this article is to apply the technique of systemism, developed by Gansen and James (2021), which uses graphics to visualize concepts and main ideas in a paper, potentially bringing forth new insights in complex fields of inquiry. Two publications that focus in varying ways on culturally immersive pedagogy will be engaged with each other via systemist visualizations for the advancement of knowledge about cross-national active learning.

## 2. Background and Agenda

While there is a call to de-colonize, tangible action in development studies is weak where it should be leading, and race, which should be included, is avoided (Patel 2020). Gorski (2008), addressing intercultural education, critiqued approaches that solely called out the issues rather than demonstrating how de-colonization is actually designed and practiced. Langdon (2013) argued that while development studies have engaged in reflexivity, the actual teaching pedagogy that instructors could follow was lacking. Recent scholarship has contributed to the imperative to center the student, de-centering the hegemonic canon (Zidani 2021), similar to a call to avoid international development industry discourse and bring in local perspectives (Cornwall 2020). Students need to understand colonial histories and be reflexive in order to unpack their own biases and perspectives (Cornwall 2020; Langdon 2013). Some generally agreed-upon imperatives include bringing diverse voices into the curriculum, teaching and utilizing reflexivity and positionality, and taking a critical look at the institutions of global development, i.e., Bretton Woods and more recent organizations. What is lacking in the current literature are pedagogical methods that have been implemented and assessed for impact, particularly culturally immersive and collaborative classroom activities, both online and in person.

This article analyzes two published papers that both focus, in some way, on culturally immersive pedagogy aimed at experiential learning activities for students to gain differing perspectives on societal needs and priorities. These papers present very different pedagogical designs and, in fact, could be considered polar opposites: one reflects on a completely online course that was nonetheless culturally immersive in some ways for the participating students from the United States and Latin America, and the other on a study abroad program that took students to Morocco.

What might these two very different courses, one virtual and one in the field, tell us about de-colonizing cross-national and cultural learning and global development studies in general through active learning? What might the similarities and differences in experience and outcomes be for an online experience versus an on-the-ground experience? And how might one article inform the other, teasing out critical aspects not otherwise noted? How does systemic notation analysis bring out new insights? A systematic synthesis of the two articles—a systemist method explained in the introduction to this special issue (Gansen and James 2023)—will be applied to answer those questions.

Systemist notation is followed in each figure that follows, and a full explanation of it appears in Gansen and James (2023). The text in each figure is typed in Upper- or lower-case characters, depending on the system level. Upper-case characters are used for macro-level variables, while lower-case characters are used for micro-level variables. Each figure also comes in double frames—the outer one refers to the environment, the inner one to the system.

The first paper (Smith et al. 2021), in Portuguese, describes a course taught online that combined students from the U.S. and Latin America in an online bilingual, collaborative research experience, seeking to surmount the barriers of language, technology, time, and cultures in an active learning context. The goal of this course was to use bilingualism, positionality, and other methods to de-colonize the hegemony of the North in academic relations across the Americas. As Spiegel et al. (2017) have noted, online courses and collaborations can re-entrench Western ideas even when offered to students in developing countries, and the use of English and digital technologies can reinforce this unequal transfer of knowledge. One of the goals of this course was to de-center English, and in fact, to

de-center language as a barrier, as students worked together to research and address pressing socioeconomic needs in their own societies. It should also be noted that most of the students did not have prior coursework or experience in research. Being new to research and bilingual collaborations, we assumed students would have fewer preconceived notions about how these processes should work, allowing them to focus instead on the societal issues they would target.

The second paper (Parmentier and Moore 2017) focuses on studying abroad in a short-term faculty-led program as an active and experiential learning activity, in particular in teaching ethics and sustainable development. The paper describes an exercise in uncovering hegemonic Western thinking, in this case, students from the U.S. using their own ethical lens to judge sustainable development in another society. The pedagogical approach was to create as many culturally immersive activities as possible where students were in Moroccan homes, interacting with Moroccan students, and participating in service projects. Assignments included journaling and photo taking to uncover the students' perspectives and assumptions.

Each paper has been translated into graphic form, as shown below, bringing out the main concepts and their interrelationships. The sections below will analyze each systemized paper and then discuss what we might learn from reflecting on them together with the visual tool to uncover key principles in de-colonizing cross-national learning in global development studies.

### 3. Systemist Analysis of the Two Papers

The graphic depiction (Figure 1 below) of paper one (Smith et al. 2021) designates Global Development Pedagogy as the system and the World Beyond as its environment. The macro and micro levels of Global Development Pedagogy correspond, respectively, to the broader rationale for de-colonizing research collaborations and the way this course was constructed, including outcomes. Analysis begins with the central problem the pedagogical project was trying to address—that local languages and knowledge are often missing from research, and thus the sources that students read and are taught to valorize are not diverse or inclusive. The pedagogical project was thus approached with a commitment to co-design the course with colleagues from the U.S. and Latin America, which included providing all reading material in English and Spanish. Another main element of de-colonizing the syllabus was to feature readings from Latin America and from many disciplines. Both faculty and students engaged in reflexive positionality exercises, examining their backgrounds and identities, how others perceive them, and how this might influence their research. The idea of symmetrical exchange is emphasized and visible in the central plain box of the graphic, and the faculty teams were conscious of this throughout the entire process of designing and teaching the course.

There were many challenges to this collaborative course, and not everything went smoothly (Smith et al. 2021, p. 74). Maintaining bilingualism was challenging, and this was a central goal of the course and a key tenet to de-colonizing the north-south collaborative teaching and research experience. Students in the course chose, in groups, pressing societal issues to research and present. Several groups chose indigenous human rights and extraction, and it became apparent that there are layers to colonization; after all, Spanish and Portuguese were colonizing languages as well. The active learning approach was present in the cross-cultural immersion that ensued in the construction of teams, which included students from the U.S. and Latin America. The presentation of findings took place through techniques of knowledge mobilization to maximize impact and societal benefit for stakeholders and policymakers. This final step illuminated the need to be aware of what knowledge we value and disseminate, as the red octagon in the graphic depicts. The cross-national nature of the collaboration made students in the US and Latin America aware of each other's histories and challenges; with Mexico, Chile, and Argentina represented, the Latin American students learned more about each other.

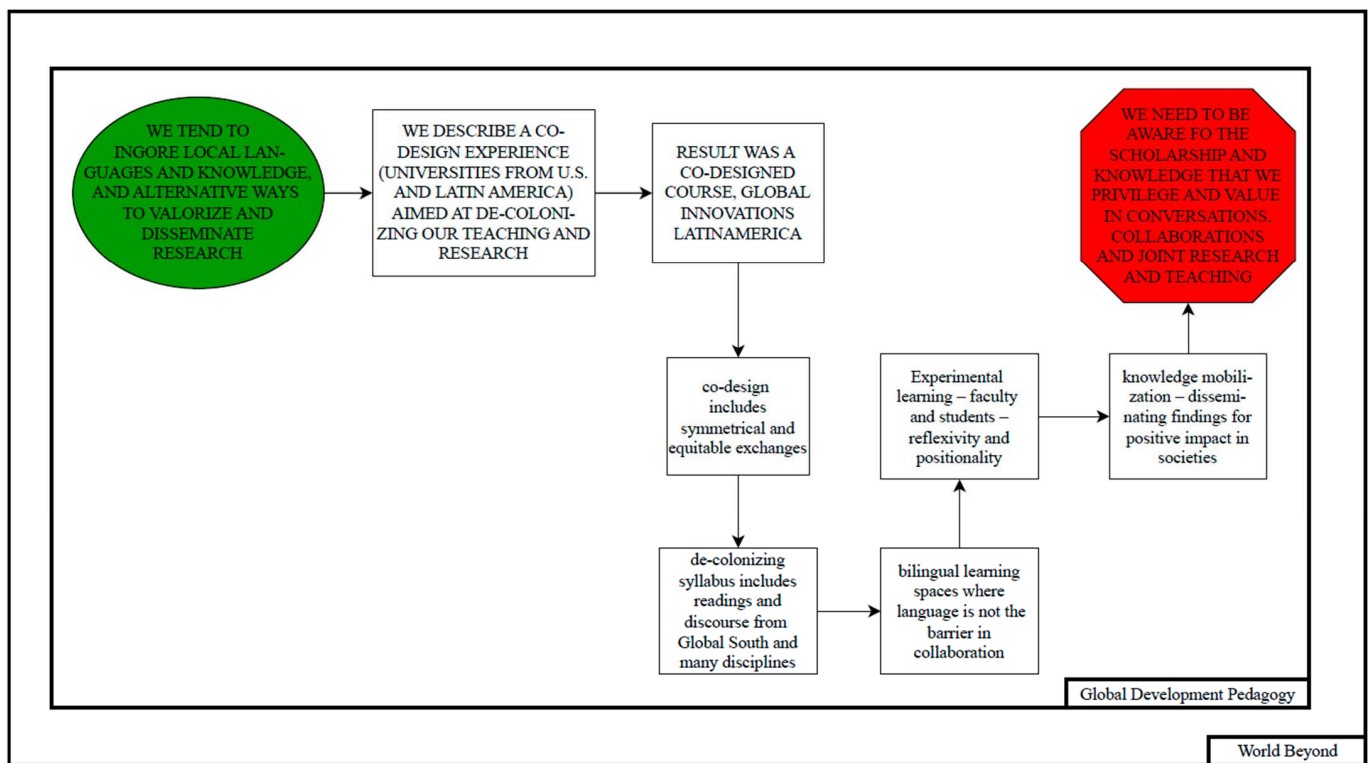

**Figure 1.** Pedagogia descolonizante Experiências CTS que descentralizam nossas ideias CTS-(De-colonizing pedagogy: Decentralizing our ideas in STS) (Smith et al. 2021) Diagrammed by: (Sarah Gansen).

Thus, as Figure 1 demonstrates, the literature and work on de-colonizing research collaborations have shown that there are biases in knowledge acquisition and valuation and even in what problems or questions are pursued in research. Cross-national teams are challenged to overcome these biases, along with language barriers that often impede these collaborations. Language and culture are intertwined, and privileging one language, such as English, creates an uneven exchange of information and knowledge. This multinational course emphasized bilingualism, a balanced share of North and South American scholarship, and knowledge mobilization to create beneficial returns for the targeted communities.

The second paper (Figure 2 below) designates the cross-cultural learning classroom as the system and the world beyond as its environment. The macro and micro levels of the cross-cultural learning classroom correspond, respectively, to the pedagogical elements of ethics and sustainable development and the processes and outcomes of the course. In many fundamental ways, this second paper is extremely different and does not explicitly purport to de-colonize global development studies. In fact, it has been argued that US study abroad programs perpetuate a form of neo colonialism and Western privilege (Bryan et al. 2022; Villarreal Sosa and Lesniewski 2021). This paper describes a program in Morocco where the US and other students had many opportunities for cultural immersion with local students and other citizens. Thus, a commonality between the projects is the immersive experience, both online and in person, for students to interact cross-culturally and nationally, albeit in very different contexts. While the immersive aspects of the program contributed to students' understanding of other contexts and priorities as well as influencing them personally to reflect on their own culturally held perspectives and values, the latter were harder to uncover.

The paper examines how studying abroad can expose students to ethical considerations when applied to sustainability and sustainable development. As Figure 2 below shows (blue parallelogram), the examination of ethical considerations that are flexible and reflect cross-cultural awareness is often missing from sustainable development pedagogy.

During the program, there were activities designed for students to reflect on their perspectives, such as the assignment to photograph something they deemed 'sustainable' and something they deemed 'unsustainable'. As the title of this paper implies, there were some surprises, as the faculty found that many students would conflate what they deemed 'good' or 'bad' with 'sustainable' or 'unsustainable'. The camel reference, for instance, was from comments made by students that camel-carrying tourists in the desert was an unsustainable activity because they felt the animals were being mistreated. Overall, there was a tendency for students to evaluate what they were observing through their own moral lenses.

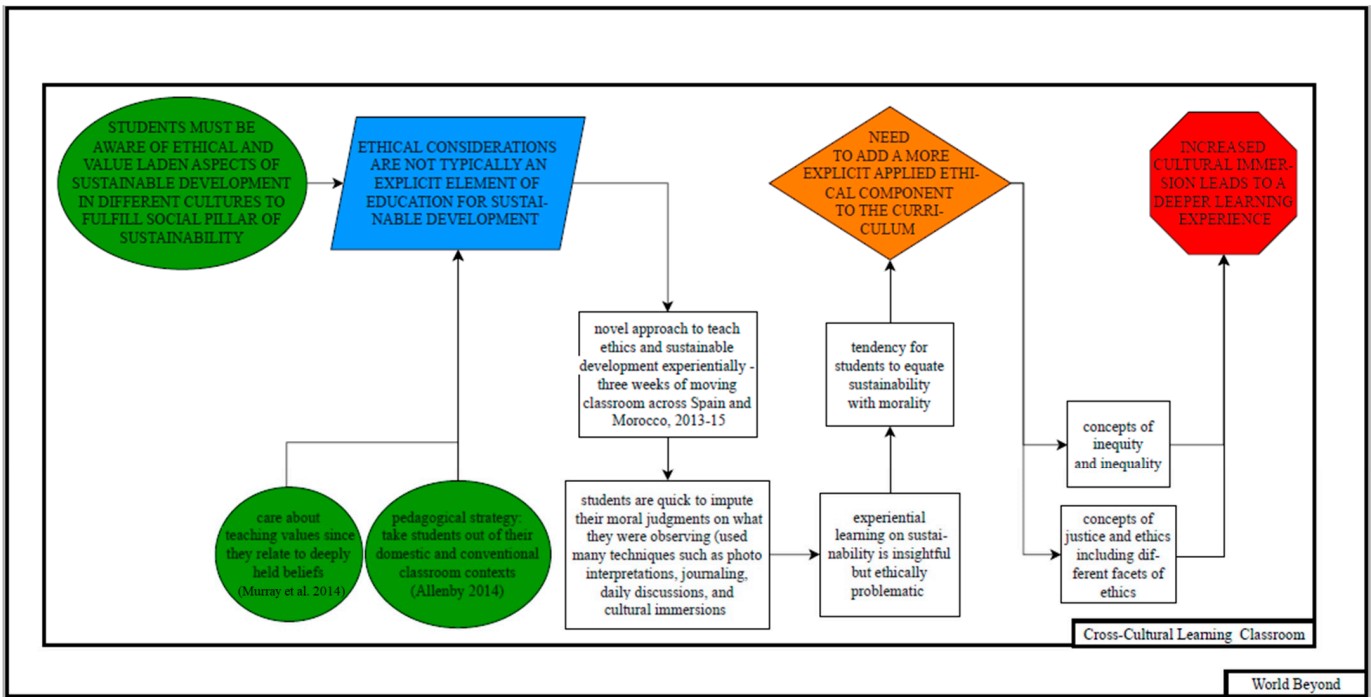

**Figure 2.** 'The Camels are Unsustainable': Using Study Abroad as a Pedagogical Tool for Teaching Ethics and Sustainable Development (Parmentier and Moore 2017; Murray et al. 2014; Allenby 2014) Diagrammed by: (Sarah Gansen).

In summary, while the reflective exercises were useful in bringing out students' perspectives on what they were observing, utilizing those reflections for a deeper cross-cultural experience was challenging (see Figure 2). Ethics training is typically not part of sustainable development studies. Students did not recognize their cognitive lenses when observing practices they deemed abhorrent, such as garbage on the side of the road or camels working to carry tourists across the desert. Despite having had some cross-cultural training before this trip, they were quick to judge rather than analyze using cultural awareness, especially when presented with images and scenes that provoked a reaction influenced by their value systems. Discussing ethics and value systems had not been a part of the pre-departure curriculum or ongoing group discussion on the program, and the conclusion indicated they should be in some way.

This, of course, is what cross-cultural learning, one of the goals of study abroad, is supposed to address. What was the problem? In considering these two papers, which describe two very different experiential learning environments and relevant literature, it is evident that cultural learning is broadly absent from research training and from sustainability studies. And this cultural learning refers to deep and intangible culture, the realm where values, mores, and ethics reside. And yet, for selecting research topics and questions and understanding issues of sustainability in different societies, these perspectives are critical for students and researchers to understand. 'De-colonizing' pedagogy that does not

take students to this level of culture will lack the power to transform and shift the lens or gaze of the participants.

On the other hand, another common element, immersion without much preparation, did result in experiential learning to some extent. However, what was noted in both instances was that self-reflection, particularly in the deeper realms of culture, was more difficult to reach. This was evident in the study abroad program, but most likely was a significant factor in the online course as well. Students were quick to use their own ethical lenses when observing behaviors in Morocco, despite having participated in cross-cultural training exercises. I would argue that in the case of online collaboration, students did the same, but there was not enough time or the right activities included to assess their cultural perspectives. Valuing different knowledge systems must begin with 'seeing' these alternative knowledge systems as well as societal priorities, and values and knowledge are often intangible and invisible to those outside the culture. This 'seeing' takes time, whether the interaction is online or in person, and many of the deep effects of colonial legacies reside in these psychological, intangible realms.

Study abroad and online collaborations offer very different forms and experiences of cross-cultural learning, but we can gain insights from considering the experiences in a comparative and contrasting way. Spiegel et al. (2017) observed that online international collaborations often manifest as top-down processes, with the courses designed in the Global North for Global South audiences, resulting in an unequal exchange of knowledge. As well, when students are studying abroad with their faculty from the US, for instance, and the curriculum is designed by those faculty, ways to collaborate with counterparts in the host country or intentionally create de-colonizing activities can easily be overlooked. In seeking insights from online collaboration, study abroad programs can be co-designed with faculty, students, and others in the host country, with activities designed to bring intercultural groups of students together for active learning. Online collaborations can benefit from more of the pre-interaction exercises implemented by study abroad programs to promote cross-cultural awareness.

## 4. Conclusions

Several significant insights were made possible by the systemic notation and analysis of both of these articles together. For more enlightened, and potentially de-colonizing active learning, the principle of co-design is significant. This could be co-designed between faculty, students, and community members where a group might visit. The co-designing method takes time, both with online collaborations and study abroad courses; it relies on building relationships and should not be rushed.

The role of culture in international collaborations that seek to de-colonize and engage students in active learning is highly significant but difficult to address and measure. The red octagons in both notations point to this—the study abroad project explicitly and the online course implicitly—in the need to consider how we value and privilege different types of knowledge. Pedagogical approaches should allow students to reflect deeply on their own cultural values, ethical guidelines, and assumptions, including what constitutes knowledge in their own disciplines and positions in society. In designing these courses and experiences, preparation is key. This means how it is done, for how long, and how the concepts and reflections are carried out through the experience.

Having co-authored both of the papers under examination and having participated in the design and teaching of both of these courses, both online and abroad, I had never considered the lessons learned from both in a co-reflexive manner. The systemist annotation technique allowed the key findings and imperatives to be illuminated more clearly, with stronger elements of de-colonizing pedagogical approaches offered for designers of these international or multinational active learning courses. Studying abroad may still be considered contradictory to de-colonizing global development studies pedagogy, but there are ways to increase learning when it is co-designed and includes cultural learning throughout. Likewise, North-South online research collaborations have the potential to

surmount barriers of language and unilateral knowledge flows, but they require careful design and the development of relationships among the students and faculty. The systemist visualization process allowed these conclusions to come to the fore when considering both projects presented in these papers.

**Funding:** This research received no external funding.

**Institutional Review Board Statement:** Not applicable.

**Informed Consent Statement:** Not applicable.

**Data Availability Statement:** Not applicable.

**Conflicts of Interest:** The author declares no conflict of interests.

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
