# Peer review of "Cross-National Active Learning in Global Development Studies: De-Colonizing the Curriculum"

_socsci, doi:10.3390/socsci12070414_

Round 1
Reviewer 1 Report
The article appears interesting and crucial, hovewer, the contents could be more elaborated in the area of comparative research concerning e.g. core curriculum in education. It could have elicited more detaled data on the necessity for transformative educational goals that are aligned with the knowledge transfer in a horisontal and vertical dimensions. The contents analyzed in the article seem to be an introductory to expansive education and expansive research. The conclusion should be more courageous and clear, also eliciting the the postulates for educational change. This can be treated as the voice calling the change within being more concrete in developing reflective skills enhancing the cognitive -humanistic approach of teacher. The education appears to be in crises, that is why, the contents are valid.
as above
Author Response
I am not sure I understand this reviewer's comments. They do not seem to be particularly applicable to my article. They mention 'expansive education and expansive research', without being clear on the meaning.
The reviewer has also said 'minor editing of English required' and yet this review provided is full of English error, and there is no suggestion on the sort of errors referred to.
I therefore am unable to respond to this reviewer.
Reviewer 2 Report
The paper adds to the current discourse around de-colonization in the context of education, and specifically decolonizing the curriculum for global development studies. It applies the technique of systemism to generate graphical representations of concepts. This is a technique new to me so I read the paper with considerable interest.
The focus is on two studies that can be seen as being polar opposites, pedagogically as one is online and the other a study abroad programme involving immersion in the ‘real’ world despite both focusing on experiential learning.
The systemist approach involves ‘translating’ concepts and ideas into a non-linguistic, graphical form. Coming from a critical thinking and problem-solving perspective, I have found that transforming and re-representing information, data and ideas into new forms to be a powerful aid to true understanding and engaging in a critical analysis. The paper does make clear how this re-representation exercise can help to reveal, identify and analyse cognitive biases and more cultural assumptions underlying our thinking (or within systems that have been heavily influenced) and so it can be seen as a powerful tool for critical thinking and in cultural studies that seek to develop an understanding of different perspectives.
All of this is very interesting and appropriate, and the conclusion shows insight and also comes with a call for more co-production methodologies and activities that cross any cultural divides rather than reinforcing top down approaches that lead to an inequal exchange and all too often force pre-existing hierarchies and biases.
Essentially the paper is an (interesting and well-written) commentary on the use of an approach rather than a research paper. I would be very interested in reading a more detailed account of how the systemist approach was applied and the process of generation the two figures – particularly in how collaboration and co-production can be used as an underlying principle in such programmes.
Author Response
Thank you for these comments.
I believe the introduction by the guest editors, on the systemic approach and process, will address your final comments.
Reviewer 3 Report
This article on active learning is far from my area of research, so I learned a lot. To me it was well written clear. I have only a few very minor grammar suggestions:
1. 18-20: grammar
22. 37-40: “t also should be noted that the majority of the literature comes from the United 38 Kingdom, Canada, and to a lesser extent, the United States.” What literature?
33. 99-100: re-write suggested
Author Response
- 18-20: grammar - Thank you for this, I corrected curriculum to curricula, and added 'with a' in the second sentence: The imperative to de-colonize universities and curricula across the globe, but particularly in former colonial powers, has proliferated and become an ongoing discourse in academia over the past decade or more. Global development studies in particular, with a close connection to international development policies and practices, has mobilized to some extent in this de-colonizing endeavor, with scholars and students participating in reflexive exercises and dialogue, questioning many past assumptions
- 37-40: “t also should be noted that the majority of the literature comes from the United 38 Kingdom, Canada, and to a lesser extent, the United States.” What literature? The literature that was previously cited; changed 'the literature' to 'this literature'.
- 99-100: re-write suggested - agreed, thank you! I now have:
"Being new to research and bilingual collaborations, we assumed students would have less pre-conceived notions about how these processes should work, allowing them to focus instead on the societal issues they would target."